# Methane Emissions and Milk Fatty Acid Profiles in Dairy Cows Fed Linseed, Measured at the Group Level in a Naturally Ventilated Housing and Individually in Respiration Chambers

**DOI:** 10.3390/ani10061091

**Published:** 2020-06-24

**Authors:** Jernej Poteko, Sabine Schrade, Kerstin Zeyer, Joachim Mohn, Michael Zaehner, Johanna O. Zeitz, Michael Kreuzer, Angela Schwarm

**Affiliations:** 1ETH Zürich, Institute of Agricultural Sciences, 8092 Zurich, Switzerland; jernej.poteko@lfl.bayern.de (J.P.); jzeitz@gmx.de (J.O.Z.); michael.kreuzer@usys.ethz.ch (M.K.); 2Agroscope, Ruminants Research Unit, 8356 Ettenhausen, Switzerland; sabine.schrade@agroscope.admin.ch (S.S.); michael.zaehner@agroscope.admin.ch (M.Z.); 3Empa, Laboratory for Air Pollution/Environmental Technology, 8600 Duebendorf, Switzerland; kerstin.zeyer@empa.ch (K.Z.); joachim.mohn@empa.ch (J.M.); 4Laboklin, Klinische Labordiagnostik, 97688 Bad Kissingen, Germany; 5Department of Animal and Aquacultural Sciences, Norwegian University of Life Sciences, P.O. Box 5003, 1432 Ås, Norway

**Keywords:** methanogenesis, methane mitigation, lipid, ruminant, cattle, emission measurement

## Abstract

**Simple Summary:**

Cows emit the greenhouse gas methane (CH_4_) as a result of microbial feed digestion. Methane emissions can be reduced by adopting nutritional strategies, such as dietary supplementation of linseed. Additionally, the oil in linseed increases the proportion of favorable fatty acids in milk fat. This study evaluated the effect of linseed on CH_4_ emission and milk fatty acid composition measured in a group of cows in a naturally ventilated barn and in individual cows in respiration chambers. The substantially higher proportions of favorable fatty acids in the milk of linseed-fed cows were detected in individual milk samples and in the milk of the herd. Therefore, the analysis of bulk milk could be a suitable control instrument for retailers. Visualizing the course of CH_4_ emissions over a whole day showed slightly lower CH_4_ values in linseed-supplemented individuals and groups. However, we found no significant reduction of CH_4_ as a result of linseed supplementation. Feed supplements in concentrations that are effective in reducing CH_4_ must show whether the reduction potential is comparable when determined at the group and individual levels.

**Abstract:**

The present study evaluated the effects of linseed supplementation on CH_4_ emission and milk fatty acid composition in dairy cows measured at the group level in an experimental dairy loose housing using a tracer gas technique and individually in tied stalls and respiration chambers. Cows (2 × 20) were maintained in two separate sections under loose-housing conditions and received a diet supplemented with extruded linseed (L) lipids (29 g·kg^−1^ dry matter) or a control (C) diet containing corn flour. Subsequently, 2 × 6 cows per dietary group were investigated in a tied-housing system and respiration chambers. Substantially higher proportions of favorable milk fatty acids were recovered in L cows when compared with C cows at the group level, making the analysis of bulk milk a suitable control instrument for retailers. Linseed supplementation resulted in a slightly lower diurnal course of CH_4_ emission intensity than the control at the group and individual levels. However, we found no more than a trend for a CH_4_ mitigating effect, unlike in other studies supplementing similar linseed lipid levels. Feed supplements in concentrations that lead to a significant reduction in CH_4_ emissions must show whether the reduction potential determined at the group and individual levels is comparable.

## 1. Introduction

Ruminants emit methane (CH_4_) as a result of enteric fermentation. Ruminal CH_4_ emissions can be mitigated by adopting several nutritional strategies [1,2]. These are advantageous over animal breeding measures as they are immediately effective [3]. In particular, dietary supplementation of crushed or extruded oilseeds has been demonstrated to mitigate enteric CH_4_ emission [4,5]. In this context, supplementing the diet with >50 g of linseed lipids per kg feed dry matter (DM) particularly decreased CH_4_ emission, but diet digestibility was concomitantly impaired [5,6]. At supplementation levels of ≤20 g linseed lipids per kg feed DM, there was no reduction in the CH_4_ emissions [7,8,9]. Thus, supplementing feed with moderate levels ranging from 26–36 g linseed lipids kg^−1^ feed DM appears to be the most promising approach to achieve a clear CH_4_ reduction with marginal effects on digestion and intake [6,10,11]. In addition, extruded linseed increases the proportion of n–3 polyunsaturated fatty acids (FAs) in milk fat, which is considered beneficial to human health (e.g., [12]). Therefore, some retailers in France and Switzerland are currently paying higher prices to farmers for milk with a higher proportion of n–3 FAs in milk fat. Label programs may require evidence of CH_4_ reduction at the herd level or changes in the FA profile of bulk milk.

Traditionally, CH_4_ emissions are measured continuously on individual animals to quantify the reduction potential of dietary interventions. Continuous measurements in respiration chambers determine total emissions, but locomotion is restrained. The sulfur hexafluoride (SF_6_) bolus technique allows free locomotion of the cows during continuous measurements [13] but detects only exhaled CH_4_. These limitations do not apply to group measurements, which concomitantly allow more natural behavior of the cows [14,15] and total emission assessment. Furthermore, climate and housing-based effects, such as slurry storage and area soiling, on CH_4_ emissions are considered. These differences could promote or reduce the effects of diet on CH_4_ emissions. However, studies at the group level in relation to dietary interventions are rare, and none of the reports studied the effect of linseed supplementation on total CH_4_ emission at the group level. Schmithausen et al. [16] quantified CH_4_ emissions from cows that were fed diets with different levels of condensed tannins in a naturally ventilated experimental dairy housing using the carbon dioxide (CO_2_) balance method. Several studies compared CH_4_ emissions obtained on individual animals in respiration chambers with those determined at the practical scale using either SF_6_ permeation tubes (e.g., [17]), the GreenFeed system (e.g., [18]) or other ventilated hoods (e.g., [13]). To our knowledge, there are currently no reported data on CH_4_ emissions for individual animals in respiration chambers vs. groups of animals in naturally ventilated dairy housing.

Therefore, this study aimed to evaluate the effect of extruded linseed supplementation on the CH_4_ emission and milk FA composition both at the group level on a practical scale in naturally ventilated housing and in respiration chambers individually. The following hypothesis was tested: moderate levels of lipids exert effects on the CH_4_ emission and milk FA composition, which are detectable and similar in extent at the individual animal and group levels. In addition, the effect on feeding behavior was investigated.

## 2. Materials and Methods 

The experiment was carried out from December 2015 to March 2016. Group level measurements were carried out in a naturally ventilated experimental loose-housing system for dairy cows at Agroscope (Waldegg, Switzerland; 47°29’22’’ N, 8°55’10’’ E, 550 m above sea level). Then, cows were measured individually at the AgroVet-Strickhof in a tied-housing barn (Wülflingen, Switzerland; 47°30’58’’ N, 8°41’49’’ E) and in respiration chambers (Eschikon, Switzerland; 47°26’53’’ N, 8°40’43’’ E). The cows were transported over a distance of 30 km from the loose- to the tied-housing and over 13 km from the tied-housing to the respiration chambers. The experimental protocol complied with the Swiss legislation on Animal Welfare and was approved (ZH091/15) by the Cantonal Veterinary Office of Zürich.

### 2.1. Animals and Experimental Design

#### 2.1.1. Measurements at the Group Level in the Loose-Housing System

For group measurements, 40 primiparous and multiparous lactating cows of Brown Swiss and Swiss Fleckvieh breeds were selected and stratified into two groups in a completely randomized manner, balanced as best as possible for lactation number, breed, body weight (BW), days in milk, and milk yield. The control (C) group consisted of 14 Brown Swiss and six Swiss Fleckvieh cows, and the linseed (L) group consisted of 11 Brown Swiss and nine Swiss Fleckvieh cows. Group level measurements were conducted over 25 days, of which 21 days were used to allow the cows to adapt to the housing and diet, and emissions were measured for four consecutive days. The adaptation period was initiated with a proportionate increase of either corn flour (in the C cows) or linseed (in the L cows) supplementation at the group level during the first 5 days at 0.2, 0.4, 0.6, 0.8, and 1.0 of the final amounts. The feed intake of the two groups was assessed at the group level, whereas feeding behavior (eating and rumination time) was determined individually for ten cows per group.

#### 2.1.2. Measurements at the Individual Level in the Tied-Housing System and in the Respiration Chambers

We randomly selected 2 × 6 exclusively multiparous cows from the 2 × 20 cows in the experimental groups for the experiments where the cows were housed in a tied-housing system. All but one cow were of the Brown Swiss breed. Individual measurements were obtained directly after the measurements at the group level. Cows were given the C and L diets (details on diet ingredients and nutrient composition are given in Section 2.2 and Table 1). The mixed ration was prepared daily at Agroscope (loose housing) and transported to the tied stall barn and respiration chambers. A period of 7 days was set to allow the cows to adapt to the tied-housing system. This was followed by a 7–day period of data and sample collection. Sampling was carried out in two blocks of 2 × 3 cows each. Within the sampling period, cows were moved in pairs, with one cow per experimental diet, into respiration chambers for an adaptation period of 6 h and gas exchange measurement period of 2 days. The tied-housing systems in the tied stall barn and in the respiration chambers were equipped with rubber mats and feces collection trays. Bedding material was omitted during sampling to avoid contamination by feces.

### 2.2. Feeding

The mixed ration consisted of grass silage, corn silage, hay, beet pulp, and concentrate (Table 1). The linseed product used (TradiLin 135; Trinova AG, Wangen, Switzerland) consisted of an extruded mixture of linseed and mill byproducts (0.6:0.4). The latter absorbs the oil released from the linseed during processing. This linseed product was added on top of the mixed ration, providing 112 g TradiLin 135 per kg of DM, equivalent to 67 g linseed and 29 g (as analyzed) of extra lipids. The cows in the C group received corn flour instead of linseed in an iso-energetic manner. The amount of linseed and corn flour was adjusted to the amount of mixed ration, ensuring that the relative intake of linseed and corn flour remained the same. In the loose-housing system, cows were fixed in the feeding barrier during the first 30 min of new feed provision to enable complete consumption of the linseed or corn flour without disturbance by dominant animals. In addition, concentrates that were rich in energy and protein were allocated according to milk yield following recommendations by Agroscope [19] by an automatic feeder (loose housing) or given manually after each feeding event (tied housing) (Table 1). Each animal received 50 g·day^−1^ of sodium chloride and 100 g·day^−1^ of a commercial mineral-vitamin mixture (Minex 976, UFA AG, Sursee, Switzerland) consisting (per kg) of 100 g calcium, 80 g phosphorus, 75 g magnesium, 20 g sodium, 4 g zinc, 3.8 g manganese, 1 g cooper, 1 g sulfur, 100 mg iodine, 40 mg cobalt, 40 mg selenium, 1,000,000 IU vitamin A, 200,000 IU.

Vitamin D_3_, 3 g vitamin E, and 100 mg biotin. New portions of diets provided with ad libitum access were offered at 16:45 after milking at the group level. Specialized equipment automatically moved feed towards the cows 18 times a day. At the individual level, feeding took place at 11:00 (10:00 in the respiration chambers), 13:00, 16:30, 23:00, and 05:30. The feed amounts offered to the two groups and to the individual cows were recorded daily. Leftovers were removed and weighed once daily. All the cows were given permanent access to water. The cows were milked at 05:30 and 16:30 at all experimental sites.

### 2.3. Measurement of Methane and Carbon Dioxide Emissions

#### 2.3.1. Emission Measurement in the Experimental Dairy Loose-Housing System (Group Level)

At the group level, emissions were measured as described in detail by Mohn et al. [20]. Briefly, the housing system provided two spatially separated experimental sections with three rows of cubicles with straw. A milking parlor and analytical devices were located between the two sections. The covered underground slurry storages were separated for each section. Management routines such as milking, feeding, and 12 dung removals per day remained the same for both groups. Each group was fed separately to avoid cross-contamination. Curtains in the facades remained closed, ensuring that wind and temperature conditions in the two sections were equivalent. Adequate ventilation was ensured by spaced boards in the upper part of the longitudinal facades. There were no other natural (e.g., ruminants) or technical emission sources near the housing. The position of the housing axis was orthogonal to prevailing wind directions, minimizing cross-contamination between the experimental sections. The dual tracer ratio method [20,21] was applied to quantify emissions for each experimental section independently and detect potential cross-contamination. It involved constant dosing of the tracer gases SF_6_ and trifluoromethyl sulfur pentafluoride (SF_5_CF_3_), one per section, at floor level using mass flow controllers (Contrec AG, Switzerland) to regulate the total flow and critical steel orifices to achieve homogenous spatial distribution. Representative air sampling in each section was accomplished with critical glass orifices (250 µm in diameter 2.5 m above the ground; Thermo-Instruments, Germany, and Louwers, The Netherlands). Concentrations of the tracer gases and target gases (CH_4_ and CO_2_) were analyzed in real time by gas chromatography with electron capture detection (GC-ECD, model 7890A, Agilent Technologies AG, Switzerland) and by cavity ring-down spectrometry (CRDS, CH_4_, CO_2_, model G2301, Picarro Inc., USA). More details on the implemented analytical technique and its performance with respect to suitability for point/areal sources, sensitivity, and uncertainty have been described in Mohn et al. [20]. The applied measurement sequence provided emission data with a temporal resolution of 10 min per section. Milking times were excluded from the analysis, as only part of the group was present in the sections during these periods.

#### 2.3.2. Emission Measurement in the Respiration Chambers (Individual Level)

The two open-circuit respiration chambers used in the present study had a volume of 19.3 m^3^ each [22]. The chambers were air-conditioned to 15 °C and 55% relative humidity at an air pressure of −60 Pa (relative to ambient). The airflow was set to 700 L·min^−1^ (Promethion FG 1000 flow generator, Sable Systems Europe GmbH, Berlin, Germany). A light:dark cycle of 16:8 was used. The CH_4_, oxygen (O_2_), and CO_2_ concentrations in the spent air were measured with a Promethion GA 4 gas analyzer (Sable Systems, Las Vegas, USA), alternating between in- and outgoing air at 1-min intervals. Calibrations of the gas analyzer were initiated automatically before each 2-day measurement period using pure (99.999%) nitrogen (N_2_) and a reference gas (19.8% O_2_, 1.0% CO_2_, 0.1% CH_4_, in N_2_). Recovery, averaging at 105%, was assessed before and after the two experimental blocks by burning propane gas. For statistical evaluation, emissions were averaged across the 2 days of measurement. The chambers were entered through the airlock for feeding, milking, and urine collection during the measurements. For feces collection, the back door was opened after 24 and 48 h of measurement.

### 2.4. Performance and Feeding Behavior (Individual Measurements in the Loose and Tied-Housing Systems)

We individually assessed the cows’ BWs after evening milking on the days before and after the 4-day measuring period in the loose housing and on the first and the last day of the 7-day sampling period in the tied-housing system. The BW was directly measured with a cattle balance (Modell FX21, Iconix, Rotorua, New Zealand) in the tied housing and estimated from girth size determinations with a measuring tape in the loose- (and tied-) housing system. The close correlation of 0.88 (n = 24, *p* < 0.001) determined between actual weights and estimates with the measuring tape allowed the application of the latter in the loose housing. During all measurement periods, milk yield was recorded individually at each milking using a milk meter (EasyFlow, Fullwood, Ellesmere, United Kingdom) in the loose housing and by weighing the buckets in the tied housing. Chewing and ruminating activity was monitored with nose band pressure sensors of the RumiWatch System equipped with the Converter V.0.7.3.36 (Itin+Hoch GmbH, Liestal, Switzerland) as described by Werner et al. [23]. This was accomplished for ten multiparous cows per group, including those subjected later to individual measurements during the 4-day measuring period at the group level and at the individual level for 5 days during the sampling period (excluding days were the cows were moved). The times of eating and ruminating and the number of chews during eating and ruminating were evaluated. In the loose-housing system, data from one cow in the C treatment group were excluded due to technical problems with the sensor.

### 2.5. Sampling

Grass silage, corn silage, and hay samples were collected at six times, i.e., at the beginning and end of the measurements in the loose- and tied-housing systems (separately in the two animal blocks). Samples of the sugar beet pulp and the three types of concentrates were taken twice during the entire study. The linseed and corn flour were sampled daily and pooled for the measurement period in the loose housing and analyzed separately for the two blocks of animals in the tied-housing system. In the loose housing, samples of leftovers were collected daily per group. In the tied-housing system, the daily feed leftovers were sampled individually and in proportion to the total amount of leftover. These samples were immediately frozen. In the tied-housing system, urine was separated from the feces by attaching urinals fixed by hook-and-loop fastener straps around the vulva of the cows. These straps were glued (ergo 5011, Kisling, Wetzikon, Switzerland) onto shaved skin. The urine was collected in a large and a small container, the latter containing 5 M sulfuric acid to avoid N losses. The total amount of feces and urine was recorded during the 7–day sampling period. Urine and feces subsamples were collected once and twice per day, respectively; samples were a constant fraction of the amounts excreted. The subsamples were frozen and later pooled for each cow. Before analysis, the feed, leftover feed, and fecal samples were dried at 60 °C until a constant mass was achieved and then ground to pass a 1-mm screen of either a cutting mill or a centrifugal mill (extruded linseed, corn flour, and concentrates). In the loose-housing system, milk samples were obtained individually at each milking on sampling days 1 and 4. A constant fraction of the yield at each milking was collected and the samples were pooled per group. In the tied-housing system, samples were collected on each sampling day from each milking, and a constant fraction of the yield at each milking was composited for each cow. Aliquots of individual and pooled milk samples were either frozen without additive or preserved with Bronopol^®^ (D&F Inc., Dublin, CA, USA).

### 2.6. Laboratory Analyses

Feeds, leftovers, and feces were analyzed according to standard procedures [24]. The DM and total ash contents (AOAC method 942.05) were determined with a thermo-gravimetric device (TGA 701, Leco Corporation, St Joseph, MI). For fiber analyses (AOAC method 973.18), a Fibertec System M (1020 Hot Extractor and 1021 Cold Extractor; Tecator, Foss Hillerød, Denmark) was used. Neutral detergent fiber (NDF) analysis was performed with the addition of heat stable α-amylase, but without sodium sulfite [25]. Acid detergent lignin was determined sequentially after acid detergent fiber (ADF) analysis by treatment with sulfuric acid (72%) for 3 h. Fiber data were corrected for ash content. Combustion energy in the feed and feces items to calculate energy intake and energy excretion with feces was quantified by a bomb calorimeter (C7000, IKA-Werke GmbH & Co. KG, Staufen, Germany). Feeds, leftovers, previously frozen milk, non-dried feces, and acidified urine were analyzed for N content (AOAC method 968.06) with a C/N analyzer (TruMac CN, Leco Corporation, St. Joseph, MI). Crude protein in the feed was defined as 6.25 × nitrogen (N). Carbon (C) content in the non-acidified urine was determined on the same C/N analyzer in order to be able to calculate urine energy (as described in Grandl et al. [26]). Ether extract (AOAC method 963.15) of feed items, leftovers, and feces was determined with a Soxhlet extraction system (model Extraktionsapparatur B-811, Büchi, Flawil, Switzerland). Bronopol-preserved milk samples were analyzed for fat, protein, lactose, and milk urea nitrogen (MUN) contents using a Fourier-transform infrared spectrophotometer (FTIR; MilkoScan FT6000, Foss, Hillerød, Denmark) at the Swiss routine milk analysis laboratory (Suisselab AG, Zollikofen, Switzerland). The MUN content was also determined in previously frozen milk by an enzymatic method [27].

For quantification of the proportions of FAs in the lipids of extruded linseed and corn flour, FAs were extracted by a solvent extractor (ASE 200, Dionex Corporation, Sunnyvale, CA) using a hexane:propane-2-ol mixture (3:2 v/v). The FAs were transformed to FA methyl esters (FAME) according to the International Union of Pure and Applied Chemistry (IUPAC) [28] method 2.301. Cleaning was performed as described in Wettstein et al. [29]. For FAME analysis, a gas chromatograph (model HP 6890 equipped with a flame ionization detector, Hewlett Packard, Palo Alto, CA, USA) and a CP7421 column (200 m × 0.25 mm, 0.25 µm; Varian Inc., Darmstadt, Germany) were used. Split injection (1:5) was applied. The internal FA standard was C11:0 (Fluka, Steinheim, Germany), and the external standard for the response factor was sunflower oil. A volume of 1 µL was injected with a constant hydrogen flow of 1.7 mL·min^−1^. The temperature program was set to 170 °C for 60 min, increased to 230 °C at a rate of 5 °C·min^−1^, ramped for 32 min, increased to 250 °C at a rate of 5 °C·min^–1^, and ramped for 15 min. For analysis of the FA in milk, samples were thawed and gently mixed to disperse milk fat. Internal standards (5 mL of *n*-heptane containing triundecanoin, tetradecenoic methylate, and trivaleranoin) were mixed with 0.5 mL of milk. Cold transesterification to FAME was done with sodium methylate [30]. The response factors obtained from C6:0, C13:0, and C19:0 triglyceride standards were used to adjust individual FA data. The same gas chromatograph and column and operational conditions were applied as for FA from feeds, except for the temperature regime which was as follows: initial temperature of 60 °C, ramped for 12 min, increased to 170 °C at the rate of 5 °C·min^−1^, ramped for 60 min, increased to 250 °C at the rate of 5 °C·min^–1^, and ramped for 20 min. The FAME was identified using a Supelco 37 component standard (Supelco Inc., Bellefonte, PA, USA). Peaks were identified using chromatograms from Collomb and Bühler [31].

### 2.7. Calculations and Statistical Analysis

In the loose- and tied-housing systems, the realized dry matter intake (DMI) of each component of the mixed ration was calculated based on the assumptions that the proportions of each component were the same in the offered feed and in the leftovers, and that the extruded linseed and corn flour were consumed completely (as confirmed by observations). One livestock unit (LU), defined as 500 kg BW [32], was used to adjust for differences in BW resulting from using only the multiparous cows in the tied stall system. The energy-corrected milk (ECM) (kg·d^–1^) was calculated as milk (kg·day^−1^) × [0.038 × fat (g·kg^−1^) + 0.024 × protein (g·kg^−1^) + 0.017 × lactose (g·kg^−1^)]/3.14. [19]. The efficiency traits calculated were milk production efficiency (ECM/BW) and feed conversion efficiency (ECM/DMI). In the loose-housing system, the individual DMI was calculated by dividing the measured total DMI of each group by the number of animals in the group (n = 20). The energy-related variables were calculated with the standard equations used in energy balance studies as listed in detail in Grandl et al. [26], where the two different methods of calculating the proportionate utilization of metabolizable energy (ME) for milk energy formation as used in the present study are described.

In the loose-housing system, the mass flow of the target gases (ṁ_target_) was calculated from the ratio between the background-corrected target gas concentration (c_target_) and tracer gas concentration (c_tracer_) multiplied with the known mass flow of the tracer gas (ṁ_tracer_), as ṁ_target_ = ṁ_tracer_ × c_target_/c_tracer_. For the loose- and tied-housing systems, the CH_4_ yield was described as the absolute CH_4_ amount in relation to the DMI, digestible organic matter (OM) intake, digestible NDF intake (dNDFI), and gross energy (GE) intake (GEI; CH_4_ conversion factor, Y_m_). The methane emission intensity was assessed by relating CH_4_ to ECM and BW. Methane emissions measured during the 2-day period in the respiration chambers under the tied-housing conditions were related to intake and ECM yield obtained across the entire 7-day sampling periods in the tied stall barn.

Data were subjected to analysis of variance to compare dietary treatments with TIBCO Spotfire+® (version 8.2 for Windows), considering diet as the fixed effect and either day (group level; [33,34]) or animal (individual level) as the experimental units. Spearman correlation coefficients were calculated for some variables. The distributional data properties were visually checked using a normal quantile-quantile plot of residuals. Significance was set at *p* < 0.05, and tendency at 0.05 ≤ *p* < 0.10. Descriptive statistics were used to evaluate the data and effects of linseed supplementation in the loose- and tied-housing systems. 

## 3. Results

### 3.1. Individual Performance, Digestibility, and Feeding Behavior (Loose- and Tied-Housing Systems)

The average BW did not differ between the C and L cows in any of the housing systems, but the average BW was approximately 80 kg higher in the subgroup of the groups used for the individual assessment (Table 2). The daily ECM and milk fat yields did not differ between C and L cows but were about 20% lower on average in the cows selected for the tied stall system compared with cows kept in the loose-housing system. Furthermore, there were no significant differences in the DM and OM intakes between the C and L cows. For C cows, the DMI was about 1 kg·day^−1^ higher at the individual level (tied housing) than at the group level (loose housing). The NDF and ADF intakes were higher (*p* < 0.05) in L cows at the group level and the ADF intake was higher at the individual level in L cows than in C cows. The NDF intake was slightly higher at the group level than at the individual level. Milk yield (per BW) and feed conversion efficiencies were not different between the L cows and C cows.

The L cows spent more time eating (min d^−1^, *p* < 0.05; min kg^−1^ DMI, *p* < 0.10) in the loose housing but the same time eating in the tied-housing system as the C cows (Table 3). The rumination time was longer for L cows than C cows (min·d^−1^, *p* < 0.10; min·kg^−1^ DMI and chews per kg^−1^ DMI, *p* < 0.01) in the tied-housing system but similarly long for L cows in the loose-housing system. The daily eating time and number of eating chews (per unit DMI) were on average lower by about 4% and the rumination time and number of ruminating chews (per unit DMI) were lower by about 18% in the tied-stall system than in the loose-housing system. Considering the eating behavior exclusively from the 2 days when cows were in the respiration chambers, the cows spent an average of 5% less time eating and ruminating, and made 9% less chews than in the entire 7-day sampling period in the tied-housing system (data not shown), and there was no difference between the two diet treatments (*p* > 0.10). The digestibility of OM, NDF, and ADF did not differ between the L and C cows at the individual level. Similarly, diet did not affect the absolute and relative N balance traits and N utilization (see Appendix A, Table A1). Only the proportion of urinary N in the total manure N tended to be greater (*p* < 0.10) for the L cows than for the C cows. Energy intake (GE, digestible energy, and ME) did not differ between diets (see Appendix A, Table A2). There was no effect of linseed on energy losses, except for the energy lost as CH_4_, which tended (*p* < 0.10) to be lower by 9% in L cows than in C cows. Concomitantly, the intake, retention, turnover, and utilization of energy did not differ between the two dietary treatment groups.

### 3.2. Methane and Carbon Dioxide Emissions (Group and Individual Level)

The carbon dioxide emissions (expressed per LU) did not differ between dietary treatments (L vs. C cows) but were on average 5% lower at the individual level than at the group level. In the loose (group) and tied (individual) housings, the CH_4_ emissions followed the typical diurnal pattern (Figure 1), with characteristic peaks after feeding (and milking) with both diets. The average CH_4_ production per animal (g·day^−1^) was 4% lower among L cows than among C cows at the group level (414 ± 5.7 vs. 431 ± 9.0) and tended (*p* < 0.10) to be lower by 9% at the individual level (447 ± 34 vs. 491 ± 40; data not shown in table). The trends for the lower CH_4_ production by L cows than the C cows were not only observed for temporal averages but also could be seen as a distinct shift in the diurnal pattern of CH_4_ emission (Figure 1). There were no significant differences in the CH_4_ traits related to intakes of DM, digestible OM or GE between the two dietary treatments (Table 3). However, the CH_4_ yield per unit of dNDFI was lower (*p* < 0.01) in the L cows than in the C cows. Methane emission intensities did not differ between the diets at both the group and individual levels. The Y_m_ and DMI were 1.1-fold higher at the individual level than the group level. The methane emission intensity per unit of ECM was 1.3-fold higher at the individual level than at the group level, whereas the CH_4_ production per LU was similar. 

### 3.3. Milk Composition (Group and Individual Level)

The milk fat and protein contents were lower (*p* < 0.01) in L cows than in the C cows both at the group (bulk milk) and individual levels (Table 4). At the group level, L cows exhibited a higher (*p* < 0.01) milk lactose content than C cows. The milk fat content of the cows selected for the tied stall system was on average slightly higher than that of all cows in the loose-housing system, but the protein and lactose contents were similar at both the individual and group levels. Enzymatic analysis revealed that MUN was significantly higher in the L cows (*p* < 0.05) than in the C cows at both the individual and group levels. The MUN analyzed by FTIR did not reveal these differences. Independent of the type of analysis, the MUN was slightly greater in the cows kept in the tied stall system than in all cows in the loose-housing system.

The milk FA composition, particularly the proportions of individual C18 FA, was substantially influenced by linseed supplementation to the diet, and this was found at both the group and individual levels. The total C18 FA proportion was higher in the L cows than in the C cows (*p* < 0.001). In particular, the C18:3 n-3 (α-linolenic acid, ALA) proportion was higher in the L cows when compared with the C cows (*p* < 0.001). The total n-3 FA proportion was higher (*p* < 0.001) and the n-6 to n-3 FA ratio was lower (*p* < 0.001) in the milk fat of L cows than that of the C cows. The milk fat of L cows had higher (mostly *p* < 0.001) proportions of the major biohydrogenation intermediates (C18:1 *trans*-11 and C18:2 *cis*-9, trans-11) and terminal products (C18:0 and C18:1 cis-9) than that of C cows. In the milk of the L than the C cows, the proportion of saturated fatty acids (SFAs) was lower and those of monounsaturated fatty acids (MUFAs) and polyunsaturated fatty acids (PUFAs) were higher (*p* < 0.001). In general, the effects of linseed feeding on the FA profile were recovered at both the group and individual levels.

## 4. Discussion

The main purpose of the present study was to investigate the extent to which a known methane-mitigating feed supplement can reduce methane emissions at both the individual level in respiration chambers and at the group level in naturally ventilated dairy housing.

### 4.1. Effects of Linseed on the Intake Pattern and Milk Yield in the Loose- and Tied-Housing Systems

In the present study, extruded linseed did not affect feed intake. The cows investigated at the individual level ate more, but this was because individual cows were the exclusively heavier multiparous cows, whereas at the group level, primiparous cows were also included. The addition of extruded linseed products in an iso-energetic manner resulted in a higher dietary content and intake of fiber. This extra fiber might explain the observed longer eating times and higher number of chews per DMI in L cows in the loose housing. In the tied-housing system, linseed also increased chews per DMI; in addition, the rumination time increased, indicating that linseed affected feeding behavior in both the loose- and tied-housing systems. Overall, rumination times were shorter in the tied-housing system than in the loose-housing system. Tied cows, which were also subjected to various experimental procedures, might have expressed less natural behavior than loose-housed cows [14,15]. The numerical increase observed in ECM in the linseed-fed cows in the loose- and tied-housing systems is in contrast with the significant increase reported by Martin et al. [6]. Despite a higher DMI, the overall ECM yield of the cows in the tied-housing system was lower than that of the cows in the loose-housing system, which was probably the result of progressing lactation, repeated transport and change in the environment (tied-housing and respiration chambers).

### 4.2. Effects of Linseed on Digestibility as well as Nitrogen and Energy Utilization (Individual Level)

At the moderate level of linseed supplementation studied, we expected only marginal effects on digestibility and utilization. We assumed that the high fiber content or the part of the lipids that remained unprotected in the linseed product could adversely affect ruminal nutrient degradation and microbial protein synthesis. In line with previous reports by Martin et al. [6] and Focant et al. [11], using 36 and 26 g linseed lipids·kg^-1^ feed DM, respectively, extruded linseed supplemented at 29 g lipids·kg^-1^ feed DM in the present study did not reduce the digestibility of fiber and organic matter. In another study, it was found that even feeding 20 to 40 g pure linseed oil·kg^−1^ DM did not affect digestibility in dairy cows [12]. Consistent with this, energy and protein retention were not affected either. Thus far, the energy and N balance of cattle fed extruded linseed have been reported for growing cattle, supplemented with only 16 g of additional linseed lipids per kg diet [7], reporting also no effect on energy and N utilization. The observed tendency towards a greater proportion of urinary N in the total manure N for the L cows than the C cows is in accordance with findings by Focant et al. [11] who reported higher urinary N proportion excreted by lactating Holstein cows fed similar linseed levels (26 g linseed lipids·kg^-1^ feed DM). The tendency to lose less energy via CH_4_ in L cows than in C cows did not lead to greater energy retention, but in the latter variable, the variation between cows was very high. The results of the present study indicate that the linseed product was indeed iso-energetic to corn flour in the present experiment. 

### 4.3. Effects of Linseed on Methane Emission at Group and Individual Level

In our study, we supplemented a grass silage based diet with a moderate level of 29 g linseed lipids per kg diet at a forage-to-concentrate ratio of 63:37. This lipid supplementation did not cause more than a trend for a CH_4_ mitigation effect, although clear effects could have been expected based on the results reported in earlier studies. For example, Martin et al. [6]), Engelke et al. [10], and Focant et al. [11] supplemented extruded linseed at a similar moderate level (26–36 g lipids·kg^−1^ diet DM) and found CH_4_ suppression by approximately 10–20%. This difference in response to extruded linseed could have resulted from the different forage type (corn vs. grass silage) and the forage-to-concentrate ratio used in the experiments. Accordingly, the effect of moderate levels of linseed on CH_4_ were weaker in the studies of Engelke et al. [10] and Martin et al. [6], when exchanging corn silage with grass silage and when reducing the concentrate proportion from 50% to 40%, respectively. This is consistent with the findings of Machmüller et al. [35], who found that using a lower forage-to-concentrate proportion led to lower methane-mitigating effects of medium-chain FA. The ruminal pH is higher in forage-based diets, which may weaken or even prevent the CH_4_-suppressing effect of linseed lipids [36]. In our study, linseed reduced the CH_4_ yield per unit of dNDFI, which could have been simply the consequence of the higher NDF intake with the linseed diet, assuming that this extra fiber from the linseed hulls and bran was not easily digestible. However, the NDF digestibility seemed to increase in linseed-fed cows, possibly due to the longer rumination time, in response to higher levels of dietary fiber [37]. When calculating correlations between the variables involved (Table 5), no relationship was found between daily rumination time and daily CH_4_ production as in Watt et al. [38]. It seems that rumination has no independent direct effect on CH_4_ production, although they are both influenced by the same factors, such as intake and fiber level [37,38,39]. Accordingly, the dry matter intake of cows was positively correlated with rumination time and ECM in the tied-housing phase. Conversely, no relationship was found between these parameters for the cows in the loose-housing phase, most likely because individual feed intake had to be calculated from group level measurements which leveled out individual differences among cows. It is noteworthy that the multiparous cows in the loose-housing phase spent more time ruminating when eating longer, but the rumination time of the same cows in the tied housing was not related to eating time. As mentioned in Section 4.1, tied cows, which were also subjected to various experimental procedures, might have expressed less natural behavior than loose-housed cows [14,15]. In addition, the observations might be explained by a compensatory relationship between eating and ruminating time [39].

### 4.4. Effects of Linseed Supplementation on Milk Composition at the Group and Individual Levels

As is known from other dietary oils, feeding cows with linseed oil in the form of extruded linseed supplementation at sufficiently high levels may decrease the milk fat and protein contents in dairy milk (e.g., Martin et al. [5]). This is caused by adverse effects on fiber digestibility mediated by the lower production of acetate, which is the major precursor of milk fat and microbial protein. Accordingly, in loose- and tied-housing systems, the milk fat and protein contents were suppressed by supplementing the cows’ diets with extruded linseed. Feeding cows extruded linseed increased the enzymatically determined MUN contents at both the group and individual levels, which was probably due to the higher crude protein content of the extruded linseed-containing diet. In the study by van Zijderveld et al. [40], a dietary mixture with extruded linseed containing less crude protein was used, which led to a slight decrease in the MUN content (estimated with a pH difference technique). Our data also show that the FTIR analysis did not result in sufficiently accurate MUN values to differentiate between dietary treatments. In this context, it is puzzling that the variation among individual animals was not higher than in the values determined enzymatically. It could be speculated that there were matrix effects that affected the FTIR measurements, such as changes in the milk fat composition of linseed-fed cows.

Linseed oil is particularly rich in the dietetically valuable C18:3 n-3. However, when fed as pure oil, a large proportion of the C18:3 n-3 is biohydrogenated by the ruminal microorganisms. This is why linseed is often provided in crushed or extruded form, as the oil in this form is at least partially rumen-protected but still able to exert some inhibiting effects on ruminal microorganisms [41]. Our results showed that part of the C18:3 n-3 was transferred intact to the milk, as its proportion was substantially elevated by linseed supplementation. However, ruminal biohydrogenation also occurred, as evidenced by elevated proportions of major biohydrogenation products such as C18:1 *trans*-11 (vaccenic acid) and C18:2 *cis*-9, trans-11 (rumenic acid; the most important conjugated linoleic acid), and C18:0 [42]. The two intermediates mentioned, but not the terminal product C18:0, are considered valuable to human health. Linseed has been reported to enhance both the n-3 FA and biohydrogenation intermediates [8,10,11]. The effects of linseed on the FA profile of milk fat in our study were similar at both the individual and group levels (bulk milk), indicating that changes can be clearly recovered in a few bulk milk samples obtained from linseed-fed herds of dairy cattle. Therefore, this method may be implemented as a control instrument by retailers who pay higher prices for milk produced from cows fed linseed.

### 4.5. Comparability of Methane Emission Measurements in an Experimental Housing System and Individually in Respiration Chambers

The methane-mitigation efficacy of extruded linseed was limited, both at the group and at individual levels. Systematic differences in CH_4_ emissions between measurements at the group and individual levels are likely at least partly attributable to indirect effects, as the loose-housed cows were exposed to transport, being tied in stalls, respiration chambers, and further experimental procedures. In addition, the subpopulation of cows that underwent measurements in respiration chambers were exclusively multiparous cows. A system-dependent difference was that CH_4_ emissions in the experimental dairy housing at the group level included housing-based sources, such as the floor’s soiling. Integrated animal and housing emissions are highly relevant from an environmental point of view [16]; however, they may mask differences between feeding treatments. In particular, Hassanat et al. [43] recently showed that CH_4_ emissions from the manure of linseed-fed cows were higher than those from the manure of cows fed a control diet devoid of linseed. In addition, certain differences in linseed intake between different group members cannot be ruled out, although this should have been minimized by fixing the cows for a certain period after the linseed supplement was provided. Simultaneous measurements performed by Schmithausen et al. [16] in a naturally ventilated experimental housing in two separate sections, similar to the setting of the present experiment, also revealed slight numerical reductions in the CH_4_ emissions in lactating cows fed a CH_4_-mitigating supplement (30 g of condensed tannins·kg^−1^ diet DM; effect reviewed in Beauchemin et al. [4]). This report illustrated that it may be possible to detect the CH_4_ reduction potential of a feed supplement in a group-level assessment. However, this is analytically more challenging, the experimental design is more complex and detecting differences in CH_4_ emissions at the group level may require a higher level of supplementation than at the individual level. In contrast to CH_4_ emissions, CO_2_ emissions were comparable at the group and individual levels. Unlike CH_4_, the majority of CO_2_ originates from nutrient oxidation in the animal’s metabolism, which is less variable than ruminal fermentation. The good comparability of CO_2_ emissions indicates that both concepts, i.e., respiration chambers and the tracer ratio technique, provide reliable results (if source strengths are comparable), whereas CH_4_ emissions at the group and individual levels might differ to some extent.

## 5. Conclusions

This is the first study to assess the CH_4_-mitigating effect of a feeding strategy evaluating measurements at the individual level in respiration chambers and measurements at the group level conducted in a naturally ventilated dairy loose-housing system. The moderate level of extruded linseed chosen only resulted in a slightly lower diurnal course of CH_4_ emission intensity than the control. A significant emission reduction was solely observed for CH_4_ per unit NDF digested. Additionally, the favorable effect of linseed on milk FA composition at both the group and individual levels suggests that no individual feed allocation and recording is necessary to control the use of linseed in diet. Overall, the results indicate that group-level measurements are more challenging with regard to the experimental conditions in the loose-housing system, but are likely suitable to detect the CH_4_ reduction potential of feed supplements provided at higher concentrations. However, appropriate analytical approaches and experimental designs are needed. Due to the higher number of influencing factors, practical measurements are relevant for estimating the effective mitigation potential in a complex barn-manure storage and removal environment.

## Figures and Tables

**Figure 1 animals-10-01091-f001:**
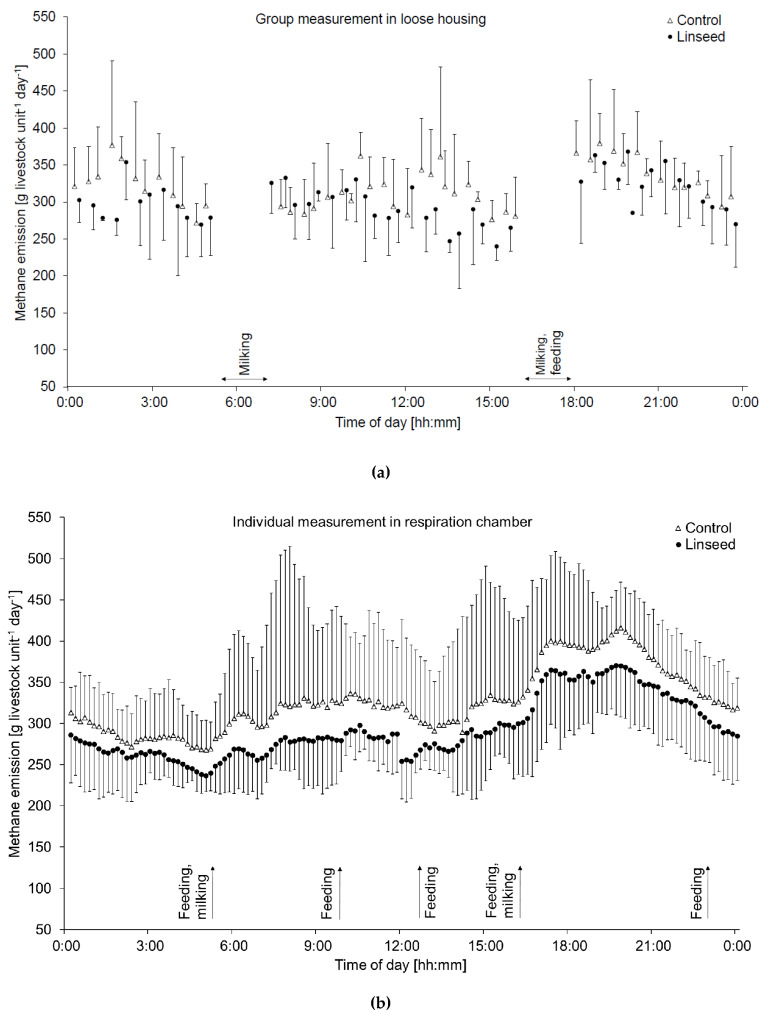
Diurnal pattern of methane emission per livestock unit (i.e., adjusted to 500 kg of body weight) of lactating dairy cows fed either a control or a linseed-supplemented diet measured (**a**) at the group level (mean of 4-day measurements in a group of 20 animals per dietary treatment) and (**b**) at the individual animal level (mean of 2-day measurements of six animals per diet). The whiskers in the graph extend to minimum (linseed) and maximum (control). The arrows indicate feeding and milking times.

**Table 1 animals-10-01091-t001:** Ingredients and their nutrient compositions and realized average intakes with the control and the extruded linseed diets^7^ in the loose and tied-housing systems (means ± standard deviation).

	Mixed Part of the Diet					Supplementary Concentrates			Linseed
Component	Grass Silage	Corn Silage	Hay	Beet Pulp	Concentrate^1^	Rich in Protein^2^	Rich in Energy^3^	Corn Flour	Product
Dry matter (g·kg^−1^)	459	417	899	260	882	863	865	859	933
**Nutrients (g·kg^−1^ of dry matter)**									
Organic matter	917	970	914	900	917	919	923	989	957
Crude protein	94	62	109	83	563	413	182	79	180
Neutral detergent fiber	463	328	506	435	221	127	161	161	398
Acid detergent fiber	303	193	295	230	135	95	85	52	215
Acid detergent lignin	37.7	33.8	37.1	29.2	50.4	32.5	32.5	22.1	103
Ether extract^4^	20.8	29.6	17.5	3.9	42.0	37.4	62.1	35.7	255
**Realized DM intake (kg·day^–1^ per cow)**									
Loose housing (group observations)^5^									
Control	6.38 ± 0.13	3.30 ± 0.07	2.32 ± 0.05	0.94 ± 0.02	1.50 ± 0.03	1.55 ± 0.53	1.05 ± 1.72	2.16 ± 0.05	-
Linseed	6.40 ± 0.14	3.31 ± 0.07	2.32 ± 0.05	0.94 ± 0.02	1.50 ± 0.03	1.73	1.14 ± 1.39	-	2.19 ± 0.05
Tied housing (individual observations)^6^									
Control	6.60 ± 0.80	3.41 ± 0.42	2.40 ± 0.29	0.97 ± 0.12	1.55 ± 0.19	1.73	1.14 ± 1.62	2.23 ± 0.27	-
Linseed	6.40 ± 0.82	3.31 ± 0.43	2.33 ± 0.30	0.94 ± 0.12	1.51 ± 0.19	1.73	1.15 ± 1.67	-	2.19 ± 0.28

^1^ Composed of soybean meal, dried distillers grains with solubles, rapeseed cake, rapeseed meal, milling byproducts, malt culms, corn gluten meal, and minerals (UFA AG, Herzogenbuchsee, Switzerland). ^2^ Composed of soybean meal, corn gluten meal, rapeseed cake, dextrose, sugar beet molasses, vitamin–mineral mixture, fruit syrup, vegetable fat, and minerals (Thurtalfutter AG, Frauenfeld, Switzerland). ^3^ Composed of barley, triticale, maize, rapeseed cake, soybean meal, peas, wheat, oat, sugar beet molasses, vegetable fat, vitamin-mineral mixture, corn gluten meal, fruit syrup, and minerals (Thurtalfutter AG, Frauenfeld, Switzerland). ^4^ Proportions of the fatty acids as analyzed (g·100 g^−1^ of total analyzed fatty acids) C16:0, C18:0, C18:1 n-9, C18:2 n-6, and C18:3 n-3 were 16.4, 1.98, 15.5, 33.0, and 24.8 in the total mixed part of the diet; 10.9, 2.18, 26.0, 54.0, and 4.51 in corn flour; and 7.04, 4.10, 16.8, 17.7, and 52.1 in linseed, respectively. ^5^ Average of group observations in the loose-housing system (two treatments × 4 days). Dry matter intake (DMI) per cow was calculated by dividing the group DMI by the number of cows. ^6^ Average of individual observations in the tied-housing system (two treatments × six animals). ^7^Contents (g/kg dry matter) of organic matter, crude protein, neutral detergent fiber, acid detergent fiber, acid detergent lignin, and ether extract were 934, 153, 351, 212, 35, and 28 for the complete control diet and 930, 165, 377, 230, 44, and 53 for the complete linseed diet, respectively.

**Table 2 animals-10-01091-t002:** Performance of cows from control and linseed treatments measured in the loose- and tied-housing phases (means ± standard deviation).

Housing	Loose (Group)			Tied (Individual)		
Item	Control	Linseed	*p*–Value	Control	Linseed	*p*–Value
Body weight (BW, kg)^1^	681 ± 93	682 ± 84	0.971	756 ± 71	761 ± 38	0.883
Energy-corrected milk (ECM, kg·day^−1^)^1^	33.2 ± 7.9	36.3 ± 10.0	0.294	26.4 ± 5.7	30.5 ± 9.3	0.386
Milk fat (kg·day^−1^)^1^	1.40 ± 0.34	1.46 ± 0.51	0.661	1.08 ± 0.21	1.16 ± 0.38	0.652
**Intake (kg·day^−1^)^2^**						
Dry matter (DMI)	19.3 ± 0.4	19.5 ± 0.4	0.417	20.1 ± 1.6	19.6 ± 2.0	0.665
Organic matter	17.9 ± 0.4	18.0 ± 0.4	0.754	18.7 ± 1.5	18.1 ± 1.9	0.564
Neutral detergent fiber	6.58 ± 0.14	7.14 ± 0.17	0.002	6.91 ± 0.29	7.24 ± 0.36	0.108
Acid detergent fiber	4.01 ± 0.09	4.40 ± 0.11	0.001	4.15 ± 0.16	4.39 ± 0.20	0.045
**Efficiency**						
ECM per BW (kg·100 kg^−1^)^1^	4.93 ± 0.12	5.37 ± 0.14	0.312	3.55 ± 0.97	4.02 ± 1.25	0.488
ECM per BW (g·kg^−0.75^)^1^	251 ± 59	273 ± 71	0.292	185 ± 48	211 ± 65	0.458
ECM^1^ per DMI^2^ (kg·kg^−1^)	1.72 ± 0.41	1.86 ± 0.51	0.111	1.31 ± 0.23	1.53 ± 0.32	0.204

^1^ Individual observations in the loose- (2 treatments × 20 animals) and tied- (2 treatments × 6 animals) housing system. ^2^ Group observations (two treatments × 4 days) in the loose housing and individual observations (2 treatments × 6 animals) in the tied-housing system.

**Table 3 animals-10-01091-t003:** Feeding behavior, digestibility, and gaseous emissions of cows from control and linseed treatments determined in the loose- and tied-housing phases (means ± standard deviation).

Housing	Loose (Group)			Tied (Individual)		
Item	Control	Linseed	*p*–Value	Control	Linseed	*p*–Value
**Eating^1^**						
min	406 ± 27	455 ± 15	0.019	422 ± 96	445 ± 36	0.354
min·kg^−1^ DMI^2^	21.1 ± 1.64	23.3 ± 0.95	0.051	20.4 ± 4.9	22.1 ± 3.0	0.219
Chews·kg^−1^ DMI	1439 ± 142	1598 ± 58	0.072	1386 ± 386	1536 ± 273	0.182
**Rumination^1^**						
min	466 ± 38	497 ± 9	0.170	411 ± 56	445 ± 34	0.078
min·kg^−1^ DMI	24.2 ± 2.2	25.4 ± 0.9	0.275	19.8 ± 2.2	22.0 ± 1.7	0.008
Chews·kg^−1^ DMI	1546 ± 146	1664 ± 66	0.162	1197 ± 172	1386 ± 188	0.006
**Digestibility (%)**						
Organic matter (OM)	-	-	-	76.1 ± 2.9	75.9 ± 3.1	0.935
Neutral detergent fiber (NDF)	-	-	-	61.5 ± 6.2	63.2 ± 4.1	0.580
Acid detergent fiber	-	-	-	63.2 ± 6.5	63.9 ± 4.2	0.829
**Carbon dioxide (kg·LU^−1^)^3,4^**	9.18 ± 0.35	9.20 ± 0.55	0.952	9.07 ± 1.02	8.54 ± 0.60	0.295
**Methane yield^4^**						
g·kg^−1^ DMI	22.3 ± 1.6	21.4 ± 2.1	0.496	24.6 ± 3.1	22.9 ± 2.1	0.285
g·kg^−1^ digestible OM intake	-	-	-	34.7 ± 3.6	32.7 ± 3.6	0.361
g·kg^−1^ digestible NDF intake	-	-	-	116.3 ± 10.0	97.9 ± 8.4	0.006
kJ·MJ^−1^ gross energy intake (Y_m_)	69.4 ± 5.0	64.1 ± 6.2	0.232	76.4 ± 9.7	68.6 ± 6.4	0.131
**Methane emission intensity^4^**						
g·kg^−1^ ECM^5^	13.6 ± 1.2	12.6 ± 1.2	0.266	19.2 ± 3.8	15.7 ± 4.4	0.165
g·LU^−1;4^	321 ± 5.1	301 ± 6.7	0.292	327 ± 44	294 ± 30	0.155

^1^ Individual observations in the loose (two treatments × 20 animals) and tied (two treatments × six animals) housing system. ^2^ Dry matter intake. ^3^ Livestock unit equivalent to 500 kg body weight. ^4^ Group observations (two treatments × 4 days) in the loose housing and individual observations (two treatments × six animals) in the tied-housing system. ^5^ Energy-corrected milk.

**Table 4 animals-10-01091-t004:** Milk composition of cows from the control and linseed treatment groups measured in the loose^1^- and tied^2^-housing phases (means ± standard deviation).

Housing	Loose (Group)^1^			Tied (Individual)^2^		
Item	Control	Linseed	*p*–Value	Control	Linseed	*p*–Value
Fat (g·kg^−1^ milk)	42.9 ± 2.5	34.3 ± 2.6	0.003	45.6 ± 6.2	36.5 ± 3.1	0.009
Protein (g·kg^−1^ milk)	37.7 ± 0.4	33.7 ± 0.3	<0.001	38.8 ± 3.5	33.1 ± 1.8	0.006
Lactose (g·kg^−1^ milk)	47.6 ± 0.4	49.1 ± 0.2	<0.001	48.1 ± 0.9	48.7 ± 0.9	0.248
**MUN (mg·100^−1^ mL milk)**						
Enzymatic	10.9 ± 0.5	13.5 ± 2.0	0.041	12.1 ± 2.0	15.6 ± 2.4	0.022
FTIR^3^	10.8 ± 1.0	10.5 ± 1.7	0.777	12.0 ± 2.1	12.8 ± 2.0	0.521
**Fatty acids (FA; g 100 g^−1^ of total FA)**						
C18:0	7.24 ± 0.07	8.89 ± 0.19	<0.001	8.99 ± 1.64	11.02 ± 1.19	0.034
C18:1 *trans*-6, *trans*-8	0.25 ± 0.01	0.56 ± 0.05	<0.001	0.34 ± 0.05	0.67 ± 0.11	<0.001
C18:1 *trans*-9	0.19 ± 0.01	0.44 ± 0.03	<0.001	0.18 ± 0.02	0.44 ± 0.06	<0.001
C18:1 *trans*-10	0.34 ± 0.00	0.75 ± 0.02	<0.001	0.42 ± 0.06	1.05 ± 0.55	0.018
C18:1 *trans*-11	0.67 ± 0.01	4.39 ± 0.27	<0.001	0.80 ± 0.29	4.37 ± 1.37	<0.001
C18:1 *trans*-12	0.045 ± 0.004	0.105 ± 0.004	<0.001	0.043 ± 0.013	0.097 ± 0.044	0.016
C18:1 *trans*-13, *trans*-14, cis-6, cis-8	0.31 ± 0.00	0.72 ± 0.04	<0.001	0.36 ± 0.04	0.83 ± 0.06	<0.001
C18:1 cis-9	16.6 ± 0.6	19.4 ± 0.4	<0.001	17.4 ± 2.1	24.0 ± 3.3	0.002
C18:1 *cis*-10	0.105 ± 0.025	0.263 ± 0.146	0.076	0.078 ± 0.017	0.137 ± 0.037	0.005
C18:1 *cis*-11	0.67 ± 0.03	1.07 ± 0.08	<0.001	0.74 ± 0.13	1.28 ± 0.12	<0.001
C18:1 *cis*-12	0.208 ± 0.010	0.514 ± 0.028	<0.001	0.259 ± 0.035	0.490 ± 0.072	<0.001
C18:1 *cis*-13	0.077 ± 0.002	0.185 ± 0.019	<0.001	0.092 ± 0.025	0.212 ± 0.019	<0.001
C18:1 *cis*-14, *trans*-16	0.17 ± 0.01	0.33 ± 0.04	<0.001	0.23 ± 0.03	0.54 ± 0.16	<0.001
C18:2 n-6 *trans*	0.11 ± 0.01	0.54 ± 0.07	<0.001	0.12 ± 0.03	0.51 ± 0.16	<0.001
C18:2 *cis*-9, *trans*-13, *trans*-8, *cis*-12	0.22 ± 0.02	0.81 ± 0.08	<0.001	0.26 ± 0.04	1.21 ± 0.30	<0.001
C18:2 *cis*-9, *trans*-12	0.068 ± 0.013	0.178 ± 0.010	<0.001	0.079 ± 0.010	0.257 ± 0.077	<0.001
C18:2 *trans*-11, *cis*-15, *trans*-9, *cis*-12	0.080 ± 0.013	1.282 ± 0.109	<0.001	0.177 ± 0.075	1.482 ± 0.349	<0.001
C18:2 n-6 *cis*	1.95 ± 0.05	1.93 ± 0.05	0.599	2.05 ± 0.23	1.92 ± 0.14	0.278
C18:2 *cis*-9, *cis*-15	0.11 ± 0.01	0.18 ± 0.01	<0.001	0.14 ± 0.02	0.20 ± 0.20	<0.001
C18:2 *cis*-9, *trans*-11	0.21 ± 0.02	0.32 ± 0.07	0.022	0.33 ± 0.07	0.36 ± 0.09	0.523
C18:2 *cis*-9, *cis*-11	0.054 ± 0.003	0.241 ± 0.016	<0.001	0.064 ± 0.014	0.250 ± 0.047	<0.001
C18:2 *trans*-9, *trans*-11	0.015 ± 0.001	0.148 ± 0.009	<0.001	0.015 ± 0.007	0.158 ± 0.046	<0.001
C18:3 n-6	0.062 ± 0.004	0.087 ± 0.012	0.008	0.065 ± 0.017	0.048 ± 0.022	0.181
C18:3 n-3	0.45 ± 0.01	1.10 ± 0.05	<0.001	0.49 ± 0.09	1.23 ± 0.06	<0.001
Σ C18 FA	30.2 ± 0.6	44.4 ± 0.8	<0.001	33.7 ± 4.0	52.8 ± 4.9	<0.001
Σ SFA	72.2 ± 0.7	60.4 ± 0.7	<0.001	70.8 ± 3.3	54.5 ± 4.7	<0.001
Σ MUFA	24.0 ± 0.6	32.4 ± 0.7	<0.001	25.0 ± 2.7	37.6 ± 4.1	<0.001
Σ PUFA	3.80 ± 0.11	7.24 ± 0.24	<0.001	4.26 ± 0.56	7.97 ± 0.79	<0.001
Σ n-6 FA	2.42 ± 0.06	2.79 ± 0.08	<0.001	2.55 ± 0.30	2.68 ± 0.14	0.368
Σ n-3 FA	0.57 ± 0.01	1.24 ± 0.04	<0.001	0.60 ± 0.11	1.35 ± 0.06	<0.001
n-6:n-3 FA	4.28 ± 0.08	2.25 ± 0.13	<0.001	4.29 ± 0.34	1.99 ± 0.07	<0.001

^1^ Group observations (two treatments × 4 days) in the loose-housing system. ^2^ Individual observations (two treatments × 6 animals) in the tied-housing system. ^3^ Fourier-transform infrared spectroscopy.

**Table 5 animals-10-01091-t005:** Correlation coefficients between performance, ruminating behavior, and methane emission of cows determined in the loose- and tied-housing phases (entire sampling period).

Trait Housing System	ECM^1^ (kg·day^−1^)	Methane^2^ (g·day^−1^)	Rumination^1^ (min·day^−1^)	Eating^1^ (min·day^−1^)
DMI (kg·day^−1^)^2^				
Loose	0.494	–0.538	–0.211	–0.107
Tied	0.777**	0.156	0.610*	–0.156
ECM (kg·day^−1^)^1^				
Loose		–0.387	0.143	0.397
Tied		0.151	0.466	0.156
Methane (g·day^−1^)^2^				
Loose			–0.322	–0.184
Tied			0.198	0.254
Rumination (min·day^−1^)^1^				
Loose				0.828*
Tied				–0.296

** *p* < 0.01; * *p* < 0.05. ^1^ Individual observations in the loose (two treatments × 20 animals) and tied (two treatments × six animals) housing system. ^2^ Group observations (two treatments × 4 days) in the loose housing and individual observations (two treatments × six animals) in the tied-housing system.

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
