# Peer review of "Methane Emissions and Milk Fatty Acid Profiles in Dairy Cows Fed Linseed, Measured at the Group Level in a Naturally Ventilated Housing and Individually in Respiration Chambers"

_animals, 2020, doi:10.3390/ani10061091_

Round 1

Reviewer 1 Report

  1. The language of this article is obscure. And the results of this article are not written accurately.
  2. In the animals and experimental design section, for group measurements, the control (C) group consisted of 14 Brown Swiss and six Swiss Fleckvieh cows, and the linseed (L) group consisted of 11 Brown Swiss and nine Swiss Fleckvieh cows. This different breed and numbers of cows in control and linseed groups would affect the test results. So this experimental design is not suitable.
  3. In the feeding section, Table 1, you should write the nutrient compositions of whole diet or TMR, not the nutrient composition of different diet of raw material. In addition, the energy index is not gross energy, it should be net energy of lactating.

Reviewer 2 Report

The paper reports a huge study comparing the effect of linseed on CH4 emission and milk fatty acid composition measured in group of cows in a naturally ventilated barn and in individual cows in respiration chambers.

The specific aim as formulated at the end of the introduction is too ambitious and in my opinion the numbers of replicates were too low “… to compare the effects of extruded linseed supplementation at group  level on a practical scale in a naturally ventilated housing with those measured individually in respiration chambers”. On the other hand, there was no statistical comparison between the two systems. Promise less, like evaluate the effect of linseed supplementation on the methane production and fatty acid composition of milk both at group level on a practical scale in a naturally ventilated housing in in respiration chambers individually.

From statistical point of view for a reliable comparison more replicates, particularly trial periods with cross over design would have been necessary. Also, there were differences that made the comparison inappropriate: no bedding in respiration chamber was used while straw was used in the loose-housing system; a total of 10 primiparous and multiparous cows were used in each treatment in loose-housing system, while 6 multiparous cows/trt were used in the individual study in respiration chamber. There was 20-25% less milk production in tied system that likely indicates that cows felt less comfortable as in group housing. Last but not least, as a matter of fact two completely different methods were applied to evaluate the methane emission in tied and loose system.

Minor comments:

It seems that the cows were transported from one institute to another one. Please, give more detail on the distance of the places, or at least give more precise location of the tied-housing barn such as of loose-housing barn.

In Table 1 add the amount of consumed TMR too.

In L202 use of measuring tape is mentioned – does it mean that BW was not directly measured in the loose-housing animals? How could the estimation error influence the presented values? That also make the comparison weaker.

Use the day as the experimental unit in 4-day-long period is not common, therefore, please add references to L303.

Please, extend the discussion in section 4.2.

The quite different correlation coefficients presented in Table 5 need more discussion.

Reviewer 3 Report

1.

At the end of the introduction, at the beginning of the discussion and at the beginning of the conclusion, it is pointed out that the main objective of the present study was to examine the extent to which linseed can reduce methane emissions, comparing measurements at individual and group level.
Unfortunately, this objective is somewhat missed out in the work. In this context it would be interesting to take a look at the CH4:CO2 ratio. This could give an indication whether similar CH4 reductions would have been determined with the CO2 balance method as with the tracer method used here.

2.

Please check the data in Table 2 and Table 4:
From the data for ECM in Table 2 and fat, protein and lactose in Table 4, the milk (kg·day-1) can be calculated using the formula from 2.7. Calculations and Statistical Analysis:

(ECM) (kg·d-1) = milk (kg·day-1) × [0.038 × fat (g·kg-1) + 0.024 × protein (g·kg-1) + 0.017 × lactose (g·kg-1)]/3.14

This milk (kg·day-1) multiplied by fat (g·kg-1) must give the milk fat (kg·day-1) shown in table 2. This applies to Tied (individual) Control: 1.08 and Linseed 1,16. But for Loose (group) Milk fat (kg·day-1) could be calculated as Control: 1.34 (instead of 1.40) and Linseed 1.33 (instead of 1.46).

Round 2

Reviewer 1 Report

The author has finished the response to the reviewer’ comments, and the quality of this article has been improved.

Reviewer 2 Report

I accept and appreciate the answers and the revisions in the manucript.